# Inconsistency in the use of the term "validation" in studies reporting the performance of deep learning algorithms in providing diagnosis from medical imaging

Dong Wook Kim[1]☯, Hye Young Jang[2]☯, Yousun Ko[3], Jung Hee Son[1], Pyeong Hwa Kim[1], Seon-Ok Kim[4], Joon Seo Lim[5], Seong Ho Park[1]*

1 Department of Radiology and Research Institute of Radiology, Asan Medical Center, University of Ulsan College of Medicine, Seoul, Republic of Korea, 2 Department of Radiology, National Cancer Center, Goyang, Republic of Korea, 3 Biomedical Research Center, Asan Institute for Life Sciences, Asan Medical Center, Seoul, Republic of Korea, 4 Department of Clinical Epidemiology and Biostatistics, Asan Medical Center, University of Ulsan College of Medicine, Seoul, Republic of Korea, 5 Scientific Publications Team, Asan Medical Center, University of Ulsan College of Medicine, Seoul, Republic of Korea

☯ These authors contributed equally to this work.
* parksh.radiology@gmail.com

**Data Availability Statement:** All relevant data are within the manuscript and its Supporting Information files.

## Abstract

### Background

The development of deep learning (DL) algorithms is a three-step process—training, tuning, and testing. Studies are inconsistent in the use of the term "validation", with some using it to refer to tuning and others testing, which hinders accurate delivery of information and may inadvertently exaggerate the performance of DL algorithms. We investigated the extent of inconsistency in usage of the term "validation" in studies on the accuracy of DL algorithms in providing diagnosis from medical imaging.

### Methods and findings

We analyzed the full texts of research papers cited in two recent systematic reviews. The papers were categorized according to whether the term "validation" was used to refer to tuning alone, both tuning and testing, or testing alone. We analyzed whether paper characteristics (i.e., journal category, field of study, year of print publication, journal impact factor [JIF], and nature of test data) were associated with the usage of the terminology using multivariable logistic regression analysis with generalized estimating equations. Of 201 papers published in 125 journals, 118 (58.7%), 9 (4.5%), and 74 (36.8%) used the term to refer to tuning alone, both tuning and testing, and testing alone, respectively. A weak association was noted between higher JIF and using the term to refer to testing (i.e., testing alone or both tuning and testing) instead of tuning alone (vs. JIF <5; JIF 5 to 10: adjusted odds ratio 2.11, $P = 0.042$; JIF >10: adjusted odds ratio 2.41, $P = 0.089$). Journal category, field of study, year of print publication, and nature of test data were not significantly associated with the terminology usage.

**Funding:** The authors received no specific funding for this work.

**Competing interests:** The authors have declared that no competing interests exist.

## Conclusions

Existing literature has a significant degree of inconsistency in using the term "validation" when referring to the steps in DL algorithm development. Efforts are needed to improve the accuracy and clarity in the terminology usage.

## Introduction

Deep learning (DL), often used almost synonymously with artificial intelligence (AI), is the most dominant type of machine learning technique at present. Numerous studies have been published on applying DL to medicine, most prominently regarding the use of DL to provide diagnoses from various medical imaging techniques [1–3]. The development of a DL algorithm for clinical use is a three-step process—training, tuning, and testing [4–6]. Of note is the difference between the second step (tuning) and the third step (testing): in the tuning step, algorithms are fine-tuned by, for example, optimizing hyperparameters; in the testing step, the accuracy of a completed fine-tuned algorithm is confirmed typically by using datasets that were held out from the training and tuning steps. Clinical adoption of a DL algorithm demands rigorous evaluation of its performance by carefully conducting the testing step, for which the use of independent external datasets that represent the target patients in real-world clinical practice is critical [3, 6–16].

Despite the notable difference between the tuning and testing steps, existing literature on DL show inconsistency in the use of the terminology "validation", with some using it for the tuning step and others for the testing step [6, 12, 17–19]. Such inconsistency in terminology usage or inaccurate use of "validation" to refer to testing are likely due to the fact that the term is typically used in general communication as well as in medicine to refer to the testing of the accuracy of a completed algorithm [6, 20], while the field of machine learning uses it as a very specific term that refers to the tuning step [4–6, 12, 17, 19, 21]. Also, the tuning step sometimes uses "cross-validation" procedures, which may create further confusion regarding the terminology for researchers who are less familiar with the methods and terms. The mixed usage of the terminology may have substantial repercussions as it hinders proper distinction between DL algorithms that were adequately tested and those that were not. The real-world performance of a DL algorithm tested on adequate external datasets tends to be lower, often by large degrees, than those obtained with internal datasets during the tuning step [3, 6, 22–24]. Therefore, such mixed usage of the terminology may inadvertently exaggerate the performance of DL algorithms to researchers and the general public alike who are not familiar with machine learning. We thus investigated the extent of inconsistency in usage of the term "validation" in studies on the accuracy of DL algorithms in providing diagnosis from medical imaging.

## Methods and materials

### Literature selection

We collected all original research papers that were cited in two recent systematic review studies [15, 18] (Fig 1). Both systematic reviews excluded studies that used medical waveform data graphics (e.g., electrocardiography) or those investigating image segmentation rather than the diagnosis and classification of diseases or disease states. The full texts of all papers collected were reviewed to confirm eligibility by four reviewers (each reviewed approximately 150 papers) (Fig 1), three of whom were medical doctors and one was a PhD; all reviewers were

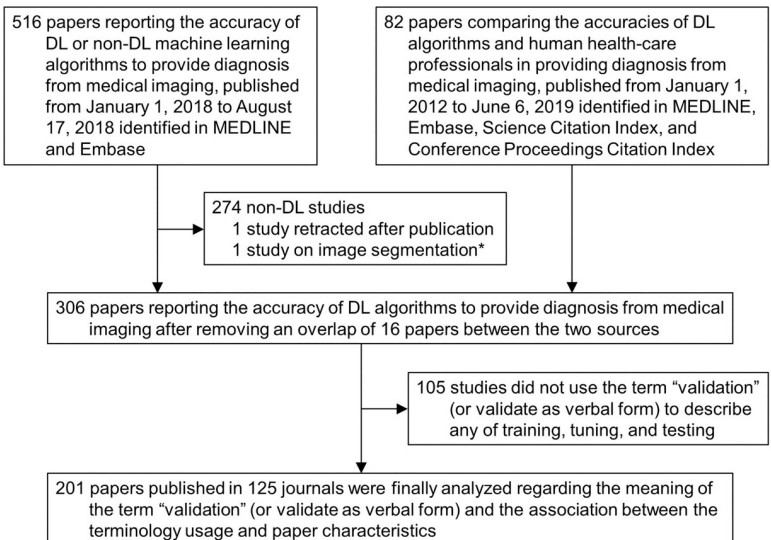

**Fig 1. Study flow diagram.** DL, deep learning. *This paper had been incorrectly characterized in the published systematic review.

familiar with studies on reporting the accuracy of machine learning algorithms as well as systematic review of literature. Prior to participation in the current study, the reviewers received reading materials [12, 15, 17, 18] and had an offline discussion to review their contents.

Specifically, one systematic review [15] searched PubMed MEDLINE and Embase to include 516 original research papers (publication date: January 1st, 2018–August 17th, 2018) on the accuracy of machine learning algorithms (including both DL and non-DL machine learning) for providing diagnosis from medical imaging. Of these, 276 papers were excluded because 274 of them were on non-DL algorithms, one was retracted following the publication of the systematic review, and another had been incorrectly characterized in the systematic review [15]. The other systematic review [18] searched Ovid-MEDLINE, Embase, Science Citation Index, and Conference Proceedings Citation Index to include 82 papers (publication date: January 1st, 2012–June 6th, 2019) that compared the accuracies of DL algorithms and human health-care professionals in providing diagnosis from medical imaging. Further details of the literature search and selection methods are described in each paper [15, 18]. After excluding a total of 16 papers that overlapped between the two systematic reviews, the reviewers checked if the term "validation" (or "validate" as a verbal form) was used in the papers to describe any of the three-step process of developing DL algorithms. As a result, 105 papers that did not use the term to describe the steps of DL algorithm development were excluded, and a total of 201 papers were deemed eligible for analysis (Fig 1).

## Data extraction

The reviewers further reviewed the eligible papers to extract information for analysis. The reviewers determined if the term "validation" (or "validate" as a verbal form) was used to indicate the tuning step alone, testing step alone, or both. We considered tuning as a step for fine-tuning a model and optimizing the hyperparameters, and testing as a step for evaluating the accuracy of a completed algorithm regardless of the nature of the test dataset used. Therefore, we did not limit the testing step to an act of checking the algorithm performance on a held-out dataset, although the use of a held-out dataset is recommended for testing (i.e., by splitting the

entire data, or more rigorously, by collecting completely external data). We then identified whether a given study used a held-out dataset for testing. "Validation" used as a part of a fixed compound term was not considered: for example, in a phrase such as "algorithm tuning used k-fold cross-validation", we did not consider this as a case of "validation" referring to the tuning step, because "cross-validation" is a fixed compound term. Papers that had ambiguity as judged by individual reviewers were re-reviewed at a group meeting involving all four reviewers and a separate reviewer who was experienced with machine learning research and 13 years of experience as a peer reviewer or an editor for scholarly journals.

In addition, the reviewers analyzed other characteristics of the papers, including the journal category (medical vs. non-medical), field of study, year of print publication, and the journal impact factor according to the Journal Citation Reports (JCR) 2018 edition if applicable. The distinction between medical vs. non-medical journals was made according to a method adopted elsewhere [15] as follows: the journals were first classified according to the JCR 2018 edition categories, and for those not included in the JCR database, we considered them as medical if the scope/aim of the journal included any fields of medicine or if the editor-in-chief was a medical doctor.

## Statistical analysis

The proportion of papers using the term "validation" (or "validate" as a verbal form) was calculated according to its usage. The overall results in all papers and the separate results broken down by the characteristics of the papers were obtained. We also analyzed whether any characteristics of the papers were associated with the usage of the terminology. For this analysis, we dichotomized the use of terminology into tuning alone vs. testing (i.e., testing alone or both tuning and testing), because we considered that using the term "validation" for meanings other than tuning, as it is specifically defined in the field of machine learning, is the source of confusion. We performed logistic regression analysis and used generalized estimating equations with exchangeable working correlation to estimate the odds ratio (OR) for using the term in the meaning of testing (i.e., OR >1 indicating a greater likelihood to use the term to refer to testing in comparison with the reference category) accounting for the correlation between papers published in the same journals. The characteristics of the papers as independent variables were used as categorical variables: journal category (medical vs. non-medical), field of study (radiology vs. others), year of print publication (before 2018, 2018, after 2018), journal impact factor (<5, 5 to 10, >10, unavailable), and nature of test data (held-out dataset vs. not held-out dataset). We combined the field of study into a binary category (radiology vs. others) because papers in individual medical disciplines other than radiology were small in number. Univariable and multivariable analyses were performed. SAS software version 9.4 (SAS Institute Inc., Cary, NC, USA) was used for statistical analysis. *P* values smaller than 0.05 were considered statistically significant.

## Results

The characteristics of the 201 papers analyzed, published in 125 journals, are summarized in Table 1 and the raw data are available as supplementary material.

Of the papers, 118 (58.7%), 9 (4.5%), and 74 (36.8%) used the term to refer to tuning alone, both tuning and testing, and testing alone, respectively. More than half of the papers used the term to specifically refer to tuning alone, which is in line with the definition used in the field of machine learning, similarly in both medical journals (97/165, 58.8%) and non-medical journals (21/36, 58.3%). Specific examples of the quotes on "validation" (or "validate") to refer to tuning and testing in the papers are shown in Table 2.

**Table 1. Characteristics of the papers.**

|  | Number of papers (%) | |
|---|---:|---|
| **Journal category** | | |
| Medical journals | 165 | (82.1) |
| Non-medical journals | 36 | (17.9) |
| **Field of study** | | |
| Radiology | 121 | (60.2) |
| Others | 80 | (39.8) |
| **Year of print publication** | | |
| Before 2018 | 10 | (5.0) |
| 2018 | 150 | (74.6) |
| After 2018 | 41 | (20.4) |
| **Journal impact factor** | | |
| <5 | 128 | (63.7) |
| 5 to 10 | 44 | (21.9) |
| >10 | 18 | (9.0) |
| Unavailable | 11 | (5.5) |
| **Nature of test data** | | |
| Held-out dataset | 133 | (66.2) |
| Not held-out dataset | 68 | (33.8) |

Table 3 shows the associations between paper characteristics and the usage of the terminology. Journal impact factors showed a weak association with the terminology usage, as papers published in journals with higher impact factors were more likely to use the term to refer to the testing step, i.e., testing alone or both tuning and testing, (vs. journal impact factor <5; journal impact factor 5 to 10: adjusted odds ratio 2.11, $P = 0.042$ with statistical significance; journal impact factor >10: adjusted odds ratio 2.41, $P = 0.089$). Journal category, field of study, year of print publication, and the nature of test data were not significantly associated with the terminology usage.

## Discussion

We found that existing literature, whether medical or non-medical, have a significant degree of inconsistency (or inaccuracy) in using the term "validation" in referring to the steps in DL algorithm development, with 58.7% of the papers using the term to refer to the tuning step alone as specifically defined in the field of machine learning. Interestingly, papers published in journals with higher impact factors were slightly more likely to use the term to refer to the testing step (i.e., testing alone or both tuning and testing).

**Table 2. Example quotes on using "validation" (or "validate" as a verbal form) to refer to tuning or testing.**

| Meaning | First author (year) | Quote |
|---|---|---|
| **Tuning** | Zhou (2018) [25] | We have randomly separated them into three parts: 400 for training, 45 for **validation** and 95 for independent test. |
| **Testing** | Bien (2018) [26] | The training set was used to optimize model parameters, the tuning set to select the best model, and the **validation** set to evaluate the model's performance. |
| | Nam (2019) [27] | The results, including AUROCs, JAFROC FOMs, and F1 scores . . . remained consistent with our results from DLAD among four external **validation** data sets. |
| | Li (2019) [28] | The high performance of the deep learning model we developed in this study was **validated** in several cohorts. |

**Table 3. Association between terminology usage and paper characteristics.**

| | Terminology usage* | | | Univariable analysis[†] | | Multivariable analysis[†] | |
|---|---|---|---|---|---|---|---|
| | Tuning alone | Both tuning and testing | Testing alone | Unadjusted OR (95% CI) | P value | Adjusted OR (95% CI) | P value |
| **Total** | 118 (58.7) | 9 (4.5) | 74 (36.8) | | | | |
| **Journal category** | | | | | | | |
| Medical journals | 97 (58.8) | 6 (3.6) | 62 (37.6) | Reference category | | Reference category | |
| Non-medical journals | 21 (58.3) | 3 (8.3) | 12 (33.3) | 1.04 (0.57, 1.89) | 0.896 | 1.22 (0.66, 2.25) | 0.528 |
| **Field of study** | | | | | | | |
| Radiology | 73 (60.3) | 5 (4.1) | 43 (35.5) | Reference category | | Reference category | |
| Other fields | 45 (56.3) | 4 (5.0) | 31 (38.8) | 1.23 (0.70, 2.16) | 0.465 | 1.05 (0.59, 1.90) | 0.862 |
| **Year of print publication** | | | | | | | |
| Before 2018 | 6 (60.0) | 0 (0.0) | 4 (40.0) | Reference category | | Reference category | |
| 2018 | 84 (56.0) | 8 (5.3) | 58 (38.7) | 1.14 (0.31, 4.24) | 0.846 | 1.36 (0.33, 5.53) | 0.667 |
| After 2018 | 28 (68.3) | 1 (2.4) | 12 (29.3) | 0.70 (0.17, 2.87) | 0.615 | 0.74 (0.18, 3.03) | 0.673 |
| **Journal impact factor** | | | | | | | |
| <5 | 82 (64.1) | 7 (5.5) | 39 (30.5) | Reference category | | Reference category | |
| 5 to 10 | 22 (50.0) | 2 (4.5) | 20 (45.5) | 1.98 (1.05, 3.73) | 0.034 | 2.11 (1.03, 4.31) | 0.042 |
| >10 | 8 (44.4) | 0 (0.0) | 10 (55.6) | 2.49 (0.98, 6.27) | 0.054 | 2.41 (0.88, 6.63) | 0.089 |
| Unavailable | 6 (54.5) | 0 (0.0) | 5 (45.5) | 1.64 (0.57, 4.76) | 0.363 | 1.57 (0.49, 5.01) | 0.447 |
| **Nature of test data** | | | | | | | |
| Held-out dataset | 73 (54.9) | 6 (4.5) | 54 (40.6) | Reference category | | Reference category | |
| Not held-out dataset | 45 (66.2) | 3 (4.4) | 20 (29.4) | 0.60 (0.33, 1.08) | 0.088 | 0.62 (0.34, 1.13) | 0.119 |

OR, odds ratio; CI, confidence interval.

*Data are numbers of papers with the % in each row category in parentheses.

[†]From logistic regression analysis with generalized estimating equations. OR >1 indicates a greater likelihood to use the term to refer to testing (i.e., testing alone or both tuning and testing) in comparison with the reference category.

Inconsistency in terminology use hinders accurate delivery of information. In this regard, some investigators advocate a uniform description of the datasets for the steps in DL algorithm development as a training set (for training the algorithm), a tuning set (for tuning hyperparameters), and a validation test set (for estimating the performance of the algorithm) [18]. However, others recommend referring to them as training, validation, and test sets [16]. "Validation" is a specific scientific term that is canonically accepted to refer to model tuning in the field of machine learning, and is also widely used in a colloquial sense to refer to testing in non-machine learning language; therefore, an attempt to enforce any one way of terminology use may likely be futile. The presence of a weak association between journal impact factor and the terminology usage (i.e., journals with higher impact factors being more likely to use "validation" to refer to testing) observed in this study should not be interpreted as providing a rationale to promote the term usage to refer to testing; rather, the data merely delineate the current pattern of term usage in the journals included in this analysis.

In order to avoid possible confusion, it would be helpful if academic journals outside the field of machine learning employ certain policy in using the term "validation" when publishing articles on machine learning, such as recommending using "validation" as a specific scientific term instead of a general word. At the very least, researchers should clarify the meaning of the term "validation" early in their manuscripts [6, 17]. As long as each paper carefully explains its definition of the term "validation", the degree and possibility of confusion would substantially decrease. A useful way for bringing the attention of researchers to terminology use and

encouraging them to use the term more accurately and clearly in their reports of machine learning research would be through guidelines for reporting research studies, most notably those set forth by the EQUATOR (Enhancing the Quality and Transparency of Health Research) Network. Specifically, a machine learning-specific version of the TRIPOD (Transparent Reporting of a multivariable prediction model for Individual Prognosis Or Diagnosis) statement, TRIPOD-ML, is currently under development [29]. Therefore, addressing the use of the term "validation" in the TRIPOD-ML would likely be an effective approach.

Another important related issue in studies reporting the accuracy of DL algorithms is the distinction between internal testing and external testing. The importance of adequate external testing using independent external datasets that represent the target patients in clinical practice cannot be overstated when testing the performance of DL algorithms for providing diagnosis [3, 6–16]. Testing with a subset split from the entire dataset, even if the subset was held out and unused for training and tuning, is not external testing and most likely insufficient [9, 30]. DL algorithms for medical diagnosis require a large quantity of data for training, and producing and annotating this magnitude of medical data is highly resource-intensive and difficult [31–34]. Therefore, the data collection process, which is mostly carried out in a retrospective manner, is prone to various selection biases, notably spectrum bias and unnatural prevalence [12, 31, 34]. Additionally, there is often substantial heterogeneity in patient characteristics, equipment, facilities, and practice pattern according to hospitals, physicians, time periods, and governmental health policies [3, 35]. These factors, combined with overfitting and strong data dependency of DL, can substantially undermine the generalizability and usability of DL algorithms for providing diagnosis in clinical practice [3, 8, 9]. Therefore, guidelines for reporting studies on DL algorithms should also instruct authors to clearly distinguish between internal testing, including the use of a held-out subset split from the entire dataset, and external testing on a completely independent dataset so as not to mislead the readers.

Our study is limited in that we could not analyze the relevant literature in its entirety. However, the two sets of papers collected from the recent systematic reviews [15, 18] may be representative of the current practice of the terminology use in DL algorithm studies, considering that the related research activity is currently most prominent in the field of medical imaging [1–3]. Also, we did not directly assess the effect of the inconsistency (or inaccuracy) in terminology usage, and the effect of mixed terminology usage on the perceived level of confusion in readers according to the field of study would be worthwhile investigating in the future.

In conclusion, our study shows the vast extent of inconsistency in the usage of the term "validation" in papers on the accuracy of DL algorithms in providing diagnosis from medical imaging. Efforts by both academic journals and researchers are needed to improve the accuracy and clarity in the terminology usage.

## Supporting information

**S1 File.**
(XLSX)

## Author Contributions

**Conceptualization:** Dong Wook Kim, Hye Young Jang, Yousun Ko, Jung Hee Son, Pyeong Hwa Kim, Seon-Ok Kim, Joon Seo Lim, Seong Ho Park.

**Data curation:** Dong Wook Kim, Hye Young Jang, Yousun Ko, Jung Hee Son, Pyeong Hwa Kim.

**Formal analysis:** Dong Wook Kim, Hye Young Jang, Yousun Ko, Jung Hee Son, Pyeong Hwa Kim, Seon-Ok Kim.

**Investigation:** Dong Wook Kim, Hye Young Jang.

**Methodology:** Dong Wook Kim, Hye Young Jang, Seon-Ok Kim, Joon Seo Lim, Seong Ho Park.

**Project administration:** Seong Ho Park.

**Resources:** Seong Ho Park.

**Software:** Seon-Ok Kim.

**Supervision:** Seong Ho Park.

**Validation:** Joon Seo Lim, Seong Ho Park.

**Writing – original draft:** Dong Wook Kim, Hye Young Jang.

**Writing – review & editing:** Yousun Ko, Jung Hee Son, Pyeong Hwa Kim, Seon-Ok Kim, Joon Seo Lim, Seong Ho Park.

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
