## [Decision Letter · Decision Letter 0]

23 Jun 2020

PONE-D-20-14331

Inconsistency in the use of the term “validation” in studies reporting the performance of deep learning algorithms in providing diagnosis from medical imaging

PLOS ONE

Dear Dr. Park,

Thank you for submitting your manuscript to PLOS ONE. After careful consideration, we feel that it has merit but does not fully meet PLOS ONE’s publication criteria as it currently stands. Therefore, we invite you to submit a revised version of the manuscript that addresses the points raised during the review process.

We look forward to receiving your revised manuscript.

Kind regards,

Julian C Hong

Academic Editor

PLOS ONE

Journal Requirements:

2.

We note that you have indicated that data from this study are available upon request. PLOS only allows data to be available upon request if there are legal or ethical restrictions on sharing data publicly. For information on unacceptable data access restrictions, please see http://journals.plos.org/plosone/s/data-availability#loc-unacceptable-data-access-restrictions.

Additional Editor Comments (if provided):

Thank you to the authors for their submission. The reviewers were positive regarding this submission with recommended revisions.

In particular, the perspectives regarding the use of the term "validation" varies across the reviewers given their respective backgrounds, and it would be helpful to discuss both viewpoints in the manuscript to accommodate the diversity of potential interested readers.

Please see reviewer responses for specific recommendations.

Reviewers' comments:

Reviewer's Responses to Questions

**Comments to the Author**

1. Is the manuscript technically sound, and do the data support the conclusions?

Reviewer #1: Yes

Reviewer #2: Yes

2. Has the statistical analysis been performed appropriately and rigorously? 

Reviewer #1: Yes

Reviewer #2: Yes

3. Have the authors made all data underlying the findings in their manuscript fully available?

Reviewer #1: No

Reviewer #2: Yes

4. Is the manuscript presented in an intelligible fashion and written in standard English?

Reviewer #1: Yes

Reviewer #2: Yes

5. Review Comments to the Author

Reviewer #1: The authors are concerned about the varying definitions of the term "validation" in medical imaging AI papers. In the machine learing world, this term usually refers to a dataset used to tune model hyperparameters and decide when to stop training optimization, but in the medical world this term usually refers to the testing of an algorithm on data not used as part of the training process at all. They show that there are many papers that use the term in the first way, and many papers that use it in the second way.

The analysis seems to be appropriate and carefully done, and the writing is very clear. In the spirit of the PLOS One data sharing policy, I think the authors need to share their raw dataset now, so the reviewers can better assess the results. There are no anonymity issues that would keep them from sharing this.

It would be helpful to include a table showing a few specific quotes from the 201 papers that illustrate the two uses of the term "validation". For readers not familiar with machine learning, this will help them understand the problem. It would also be interesting to know if all 201 papers included a separate held-out test set, or if some never tested their model on a held-out set (which would be bad).

Since it is unlikely the use of the term "validation set" to describe a tuning set will change in the machine learning world, I do not not think medical AI papers will ever have completely standardized terminology. As long as each paper carefully explains its definitions, I do not think it is a major issue. So I am not completely on board with the authors' conclusions about the need for strict guidelines on this topic.

Reviewer #2: This paper provides a systematic review and meta-analysis of the term ‘validation’ as used in deep learning studies regarding medical diagnosis. There is a major inconsistency in the medical imaging community regarding the correct use of ‘validation’ in the technical sense. This paper is well written and the topic is important to readers of both PLoS One and the medical imaging community. However, there are several aspects of this paper that I feel need to be addressed prior to publication. My general comments are as follows.

First, I don’t believe this is an inconsistency problem (which implies that there is a lack of standard terminology), but rather an inaccuracy problem regarding the correct use of a well-defined, technical term. As the authors correctly point out, the inconsistency of the term ‘validation’ is most likely because the term is used in general communication to refer to testing the accuracy of a completed algorithm. However, the term means something very specific when referring to the science of machine learning, where it is used to define the tuning of hyper-parameters in a model (i.e., validation the training procedure, not the generalization capacity). This definition is canonically accepted, and in general, the term ‘validation’ should not be mis-used in the colloquial sense when referring to technical work. In my opinion, we simply need to educate the medicine community to use the appropriate terminology when applying machine learning approaches, and this confusion won’t be a problem.

As such, I do not agree with lines 204-206, where the authors argue, “It would be ideal if medical journals would unify the use of the term validation to refer to to the testing step instead of the tuning step”. In my opinion, there is nothing to ‘unify’ here, and instead the medical journals need to enforce the correct use of terminology. If research related to machine learning (including deep learning) is to be published in scientific journals (including imaging journals), it should use canonical terminology. Superficially changing these technical terms to colloquial terms would only cause more confusion, and question sound scientific reasoning published in the journals.

Next, while I find it interesting that papers published in high impact factor journals tend to use ‘validation’ to refer to model testing (instead of the more appropriate and accurate model tuning), I don’t think this statistic should be used as a rationale to justify the proposal in Lines 206-207 as stated above. If anything, it makes more sense to test if there was a statistically significant difference in use of ‘validation’ between technical machine learning journals and imaging journals, regardless of impact factor.

The authors argue at Lines 206-207, “Such a unification in terminology use may be difficult in disciplines such as machine learning, where the term is relatively widely used to refer to the tuning step”. This line similarly does not make much sense to me; machine learning is the discipline being used, regardless of the application (i.e., medical imaging or otherwise). In the context of this paper, it is all machine learning, and imaging is simply the application. It is not a one-to-one fair comparison.

On Lines 219-236, the authors discuss “internal validation vs. external validation”. Here, they are using the term in the colloquial sense, but in the paragraphs that follow, they actually explain a process that is technically already defined as “internal testing vs. external testing”. By definition, a model that has been validated on an internal dataset (i.e., the learning procedure has been (cross)-validated to identify optimal hyper-parameters) has to be tested on a separate dataset not included in the original training and validation procedure, either internally (on a subset of the dataset held out) or externally (on a completely independent dataset). The notion of “internal validation vs. external validation” is ill-defined and goes against the entire purpose of this paper.

In technical machine learning workflows, it is often implied that model validation is based on some form of a cross-validation procedure, where the the training of a machine learning algorithm is applied to sub-sets (e.g., folds) of the data set to best learn the optimal hyper parameters. The authors to not mention this at all, and it is important as it may help to better define and impairment proper terminology.

Finally, I think that the Discussion section needs to be substantially expanded on, to include specific examples that are related to imaging. Further, I would like to see the authors also take more of a stance on promoting the appropriate use of the term ‘validation’, as stated in the above points. While the Discussion section does a nice job at documenting the different uses of the term ‘validation’, in my opinion it falls short in promoting a good solution, and should focus primarily on well-defined definitions as a reference point.

6. PLOS authors have the option to publish the peer review history of their article (what does this mean?). If published, this will include your full peer review and any attached files.

Reviewer #1: Yes: Michael Gensheimer

Reviewer #2: No

---

## [Author Response · Author response to Decision Letter 0]

23 Jul 2020

Reviewer #1: 

The authors are concerned about the varying definitions of the term "validation" in medical imaging AI papers. In the machine learing world, this term usually refers to a dataset used to tune model hyperparameters and decide when to stop training optimization, but in the medical world this term usually refers to the testing of an algorithm on data not used as part of the training process at all. They show that there are many papers that use the term in the first way, and many papers that use it in the second way.

1. The analysis seems to be appropriate and carefully done, and the writing is very clear. In the spirit of the PLOS One data sharing policy, I think the authors need to share their raw dataset now, so the reviewers can better assess the results. There are no anonymity issues that would keep them from sharing this.

Author response: Thank you for reviewing our study in detail and providing critical comments. We fully agree with your comment that the raw dataset should be shared. We have included the raw data as supplementary material.

2. It would be helpful to include a table showing a few specific quotes from the 201 papers that illustrate the two uses of the term "validation". For readers not familiar with machine learning, this will help them understand the problem. It would also be interesting to know if all 201 papers included a separate held-out test set, or if some never tested their model on a held-out set (which would be bad).

Author response: This is a great suggestion. As per your suggestion, we have added Table 2 to provide example quotes from the papers regarding the terminology usage. 

Also, regarding the use of the held-out test set, a small fraction of the 201 papers did not use a held-out dataset for testing and instead evaluated the performance of a complete algorithm using cross-validation methods. We considered the testing step as a stage for assessing the accuracy of a completed algorithm regardless of the nature of the test dataset used. Therefore, we did not limit the testing step to the act of checking the algorithm performance on a held-out dataset, even though the use of a held-out dataset is recommended for testing. 

As this new information has been added, we have also revised the statistical analysis section to explore if the use of a held-out test set was associated with the terminology usage. As a result, we found that there was no significant difference between studies that used a held-out test set and those that did not. We added these issues in the revised manuscript, which are indicated by the marginal memos reading “R1-2”.

3. Since it is unlikely the use of the term "validation set" to describe a tuning set will change in the machine learning world, I do not not think medical AI papers will ever have completely standardized terminology. As long as each paper carefully explains its definitions, I do not think it is a major issue. So I am not completely on board with the authors' conclusions about the need for strict guidelines on this topic.

Author response: We agree with your comment. We have revised the Discussion section according to the reviewer’s opinion (marginal memos “R1-3”). 

Thank you again for reviewing our study in detail and providing critically helpful comments. We hope that our responses and revisions are satisfactory.

Reviewer #2: 

This paper provides a systematic review and meta-analysis of the term ‘validation’ as used in deep learning studies regarding medical diagnosis. There is a major inconsistency in the medical imaging community regarding the correct use of ‘validation’ in the technical sense. This paper is well written and the topic is important to readers of both PLoS One and the medical imaging community. However, there are several aspects of this paper that I feel need to be addressed prior to publication. My general comments are as follows.

1. First, I don’t believe this is an inconsistency problem (which implies that there is a lack of standard terminology), but rather an inaccuracy problem regarding the correct use of a well-defined, technical term. As the authors correctly point out, the inconsistency of the term ‘validation’ is most likely because the term is used in general communication to refer to testing the accuracy of a completed algorithm. However, the term means something very specific when referring to the science of machine learning, where it is used to define the tuning of hyper-parameters in a model (i.e., validation the training procedure, not the generalization capacity). This definition is canonically accepted, and in general, the term ‘validation’ should not be mis-used in the colloquial sense when referring to technical work. In my opinion, we simply need to educate the medicine community to use the appropriate terminology when applying machine learning approaches, and this confusion won’t be a problem.

Author response: Thank you for reviewing our study in detail and providing critical comments. We agree with your comment, and we revised the manuscript accordingly. Also, considering the reviewer’s opinion, we also thought that it would be more reasonable to consider that using the term “validation” for meanings other than tuning, as it is specifically defined in the field of machine learning, is the source of confusion. Therefore, we revised the statistical analysis using the revised binary categorization of tuning alone vs. testing (testing alone or both tuning and testing) instead of the previous categorization of testing alone vs. tuning (tuning alone or both tuning and testing). 

The OR was thus re-calculated for using the term to refer to testing (i.e., OR >1 indicating a greater likelihood to use the term to refer to testing in comparison with the reference category), whereas the same was calculated for using the term to refer to tuning in the original version of the manuscript. The updated statistical results showed consistent results with the previous version, although the exact numerical values have been changed. Please refer to the corresponding revised portions of the manuscript, which are indicated by the marginal memos reading “R2-1”.

2. As such, I do not agree with lines 204-206, where the authors argue, “It would be ideal if medical journals would unify the use of the term validation to refer to to the testing step instead of the tuning step”. In my opinion, there is nothing to ‘unify’ here, and instead the medical journals need to enforce the correct use of terminology. If research related to machine learning (including deep learning) is to be published in scientific journals (including imaging journals), it should use canonical terminology. Superficially changing these technical terms to colloquial terms would only cause more confusion, and question sound scientific reasoning published in the journals.

Author response: We agree with your comment on how the medical community should be more aware of the correct use of said terminology. We have revised a paragraph in the Discussion section accordingly (marginal memo “R2-2”).

3. Next, while I find it interesting that papers published in high impact factor journals tend to use ‘validation’ to refer to model testing (instead of the more appropriate and accurate model tuning), I don’t think this statistic should be used as a rationale to justify the proposal in Lines 206-207 as stated above. If anything, it makes more sense to test if there was a statistically significant difference in use of ‘validation’ between technical machine learning journals and imaging journals, regardless of impact factor.

Author response: We agree that the statistical results should not be used as a rationale to justify the proposal as stated above. We have revised the Discussion accordingly (marginal memos “R2-3”).

4. The authors argue at Lines 206-207, “Such a unification in terminology use may be difficult in disciplines such as machine learning, where the term is relatively widely used to refer to the tuning step”. This line similarly does not make much sense to me; machine learning is the discipline being used, regardless of the application (i.e., medical imaging or otherwise). In the context of this paper, it is all machine learning, and imaging is simply the application. It is not a one-to-one fair comparison.

Author response: We agree with your comment. We have revised the Discussion section accordingly (marginal memo “R2-4”).

5. On Lines 219-236, the authors discuss “internal validation vs. external validation”. Here, they are using the term in the colloquial sense, but in the paragraphs that follow, they actually explain a process that is technically already defined as “internal testing vs. external testing”. By definition, a model that has been validated on an internal dataset (i.e., the learning procedure has been (cross)-validated to identify optimal hyper-parameters) has to be tested on a separate dataset not included in the original training and validation procedure, either internally (on a subset of the dataset held out) or externally (on a completely independent dataset). The notion of “internal validation vs. external validation” is ill-defined and goes against the entire purpose of this paper.

Author response: As per your comment, we have revised the terms to “internal testing” and “external testing”. Thank you for the keen comment.

6. In technical machine learning workflows, it is often implied that model validation is based on some form of a cross-validation procedure, where the the training of a machine learning algorithm is applied to sub-sets (e.g., folds) of the data set to best learn the optimal hyper parameters. The authors to not mention this at all, and it is important as it may help to better define and impairment proper terminology.

Author response: We have revised the corresponding portions as suggested.

7. Finally, I think that the Discussion section needs to be substantially expanded on, to include specific examples that are related to imaging. Further, I would like to see the authors also take more of a stance on promoting the appropriate use of the term ‘validation’, as stated in the above points. While the Discussion section does a nice job at documenting the different uses of the term ‘validation’, in my opinion it falls short in promoting a good solution, and should focus primarily on well-defined definitions as a reference point.

Author response: We have newly added Table 2 to show the specific examples from the reference papers, and revised the Discussion in multiple locations to incorporate the points raised by the reviewer (marginal memo “R2-7”). 

Again, we very much appreciate your insightful inputs. We hope our revisions are satisfactory.

---

## [Decision Letter · Decision Letter 1]

27 Aug 2020

Inconsistency in the use of the term “validation” in studies reporting the performance of deep learning algorithms in providing diagnosis from medical imaging

PONE-D-20-14331R1

Dear Dr. Park,

We’re pleased to inform you that your manuscript has been judged scientifically suitable for publication and will be formally accepted for publication once it meets all outstanding technical requirements.

Kind regards,

Julian C Hong

Academic Editor

PLOS ONE

Additional Editor Comments (optional):

Thank you to the authors for their work and revisions. There are minor recommendations for additional citations to include in the manuscript.

Reviewers' comments:

Reviewer's Responses to Questions

**Comments to the Author**

1. If the authors have adequately addressed your comments raised in a previous round of review and you feel that this manuscript is now acceptable for publication, you may indicate that here to bypass the “Comments to the Author” section, enter your conflict of interest statement in the “Confidential to Editor” section, and submit your "Accept" recommendation.

Reviewer #1: All comments have been addressed

Reviewer #2: All comments have been addressed

2. Is the manuscript technically sound, and do the data support the conclusions?

Reviewer #1: (No Response)

Reviewer #2: Yes

3. Has the statistical analysis been performed appropriately and rigorously? 

Reviewer #1: (No Response)

Reviewer #2: Yes

4. Have the authors made all data underlying the findings in their manuscript fully available?

Reviewer #1: (No Response)

Reviewer #2: Yes

5. Is the manuscript presented in an intelligible fashion and written in standard English?

Reviewer #1: (No Response)

Reviewer #2: Yes

6. Review Comments to the Author

Reviewer #1: (No Response)

Reviewer #2: In the Introduction, please include some references when discussing cross-validation, and how it fits into a training/cross validation + testing experimental design. This paper is a good example of a good deep learning implementation:

C. Wang, et. al. Dose-Distribution-Driven PET Image-based Outcome Prediction (DDD-PIOP): A Deep Learning Study for Oropharyngeal Cancer IMRT Application. Frontiers in Oncology. 2020. https://doi.org/10.3389/fonc.2020.01592

Also in the introduction please add cite example references when discussing the fine-tuning of a model during the validation phase. This paper is a good example of a network that was tuned during validation, using the correct terminology very well:

Y. Chang, et. al. Development of realistic multi-contrast textured XCAT (MT-XCAT) phantoms using a dual-discriminator conditional-generative adversarial network (D-CGAN). 2020. Physics in Medicine and Biology. 2020 Mar;65(6).

7. PLOS authors have the option to publish the peer review history of their article (what does this mean?). If published, this will include your full peer review and any attached files.

Reviewer #1: **Yes: **Michael F. Gensheimer

Reviewer #2: No

---

## [Editor Report · Acceptance letter]

1 Sep 2020

PONE-D-20-14331R1 

Inconsistency in the use of the term “validation” in studies reporting the performance of deep learning algorithms in providing diagnosis from medical imaging 

Dear Dr. Park:

I'm pleased to inform you that your manuscript has been deemed suitable for publication in PLOS ONE. Congratulations! Your manuscript is now with our production department. 

Kind regards, 

on behalf of

Dr. Julian C Hong 

Academic Editor

PLOS ONE